# In-flight Wind Field Identification and Prediction of Parafoil Systems

**Haitao Gao [1], Jin Tao [2,3,*], Matthias Dehmer [4,5], Frank Emmert-Streib [6,7], Qinglin Sun [5], Zengqiang Chen [5], Guangming Xie [3] and Quan Zhou [2]**

1   College of Electrical and Electronic Engineering, Anhui Science and Technology University, Bengbu 233030, China; gc_0532@126.com
2   Department of Electrical Engineering and Automation, Aalto University, 02150 Espoo, Finland; quan.zhou@aalto.fi
3   College of Engineering, Peking University, Beijing 100871, China; xiegming@pku.edu.cn
4   University of Applied Sciences Upper Austria, Campus Steyr, Wehrgrabengasse 1, 4040 Steyr, Austria; Matthias.Dehmer@umit.at
5   College of Artificial Intelligence, Nankai University, Tianjin 300071, China; sunql@nankai.edu.cn (Q.S.); chenzq@nankai.edu.cn (Z.C.)
6   Predictive Society and Data Analytics Lab, Faculty of Information Technology and Communication Sciences, Tampere University, 33100 Tampere, Finland; v@bio-complexity.com
7   Institute of Biosciences and Medical Technology, 33520 Tampere, Finland
*   Correspondence: jin.tao@aalto.fi; Tel.: +358-44-982-8138

**Abstract:** The wind field is an essential factor that affects accurate homing and flare landing of parafoil systems. In order to obtain the ambient wind field during the descent of a parafoil system, a combination method of in-flight wind field identification and prediction is proposed. First, a wind identification method only using global position system information is derived based on the flight dynamics of parafoil systems. Then a wind field prediction model is constructed using the atmospheric dynamics, and the low-altitude wind field is predicted based on the identified wind field of high-altitude. Finally, simulations of wind field identification and prediction are conducted. The results demonstrate that the proposed method can identify the wind fields precisely and also predict the wind fields reasonably. This method can potentially be applied in practical parafoil systems to provide wind field information for homing tasks.

**Keywords:** wind field; identification; prediction; parafoil system; autonomous homing

## 1. Introduction

A parafoil system is a type of unmanned autonomous vehicle, which possesses the advantages of exceptional controllability and gliding performance, and can achieve fixed-point autonomous homing and soft landing compared with traditional parachute systems [1–4]. Parafoil systems play a significant role and have wide applications in the area of disaster assistance, aerospace, and military fields [5–8]. Due to their slow airspeed in comparison to conventional fixed-wing aircrafts, the parafoil systems are extremely susceptible to wind fields [9,10]. In particular, disturbed by both the speed and direction of the ambient wind field, the parafoil system easily deviates from its desired flight path and generates position drifts, even leading to a stall [1,11,12]. Accurately obtaining the ambient wind field of the parafoil system can improve the performance of homing trajectory planning and controlling, which is a key to achieve precise autonomous homing of parafoil systems [13–15].

Researchers have explored a variety of methods to obtain wind field information with certain success. Cacan et al. [16] reported on a ground-based mechatronic system consisting of a cup and

vane anemometer coupled to a guided airdrop system through a wireless transceiver, which can provide an improved, real-time estimate of the wind profile. Wu [17] calculated the ambient wind field using the velocity data measured by the velocity sensor and the position information of the parafoil. However, the velocity measurement sensor is hard to install properly in practice because of the flexibility of the parafoil canopy [18]. Herrmann et al. [19] sampled the wind field by using a ground laser radar system to obtain the ambient wind field data of the landing point, which was sent to the control system to amend the flight. This method can measure the wind data accurately, but its implementation is relatively complicated and expensive, and it is not suitable in areas where ground laser radars are unavailable or not allowed. Kelly and Pena [20] provided a technique for obtaining accurate wind estimates in a pre-flight airdrop area using a global positioning system (GPS) dropsonde. However, due to the high uncertainty and variability of wind fields, the estimation results may not satisfy actual airdrop requirements. Besides the direct measurement of the wind field, researchers have also attempted to estimate the wind field using online estimation from its motion and trajectories. Altmann [21] proposed enhanced guidance, navigation, and control strategy that combines pre-flight wind information with in-flight identification for maximum resistance against winds and precise landing. Rogers and Slegers [22] developed a guidance strategy that uses massively parallel Monte Carlo simulation to rank candidate trajectories in terms of robustness to wind uncertainty. Hunter and Nathan [23] applied wavelet generation to generate a time series model for wind prediction, but the time range is a few days, not a few minutes after the parafoil system launches, which cannot rapidly obtain real-time wind information may result in imprecise landing. Rodriguez et al. [24] presented a system based on small unmanned aerial systems for the identification of wind features, such as gusts and wind shear. Neummann and Bartholmai [25] used a quadrotor Unmanned Autonomous Vehicle system for wind speed estimation based primarily on the wind speed triangle and the vector difference between the ground speed and estimated speed. However, none of them could acquire altitude-dependent wind field information in a range of altitude.

Despite all the achievements, low-cost methods suitable for estimating the wind field of parafoil systems is still lacking. The traditional wind field acquisition methods mentioned above are complicated and expensive to implement. The state-of-the-art motion-based wind field estimation methods are limited to estimate the wind field of a certain altitude. The wind field, however, varies with the altitude. Therefore, it is desirable to develop a novel online wind field estimation method for parafoil systems using only the GPS information during the flight, which can estimate the wind field at different altitudes and can be achieved with reduced cost and complexity compared to traditional methods.

In this paper, we study a combination method of in-flight wind field identification and prediction for parafoil systems. Firstly, a wind field identification method is derived for online wind field identification. Then, we derive a wind field prediction model based on the atmospheric motion during an airdrop. Combined with the wind field identification results, the proposed prediction model is used to predict the altitude dependent wind fields in lower altitudes. Finally, the effectiveness of the proposed methods is verified by different simulation cases.

The remainder of this paper is organized as the following. In Section 2, a dynamic model of parafoil systems is given briefly. A wind field identification method using only GPS information is discussed in Section 3. In Section 4, a wind field prediction model and an average wind field identification-prediction method are presented. Section 5 discusses the simulation results before we conclude the paper by Section 6.

## 2. Parafoil System Model

A schematic of parafoil system is shown in Figure 1. Due to the materials of the suspension lines and the connecting belt are usually high in strength and small in deformation so that the elastic deformation of the ropes can be neglected, thereby limiting the relative rolling motion between the

parafoil and the payload. Considering the relative pitch and relative yaw motions between two bodies, an eight-degree-of-freedom dynamic model of parafoil systems is established.

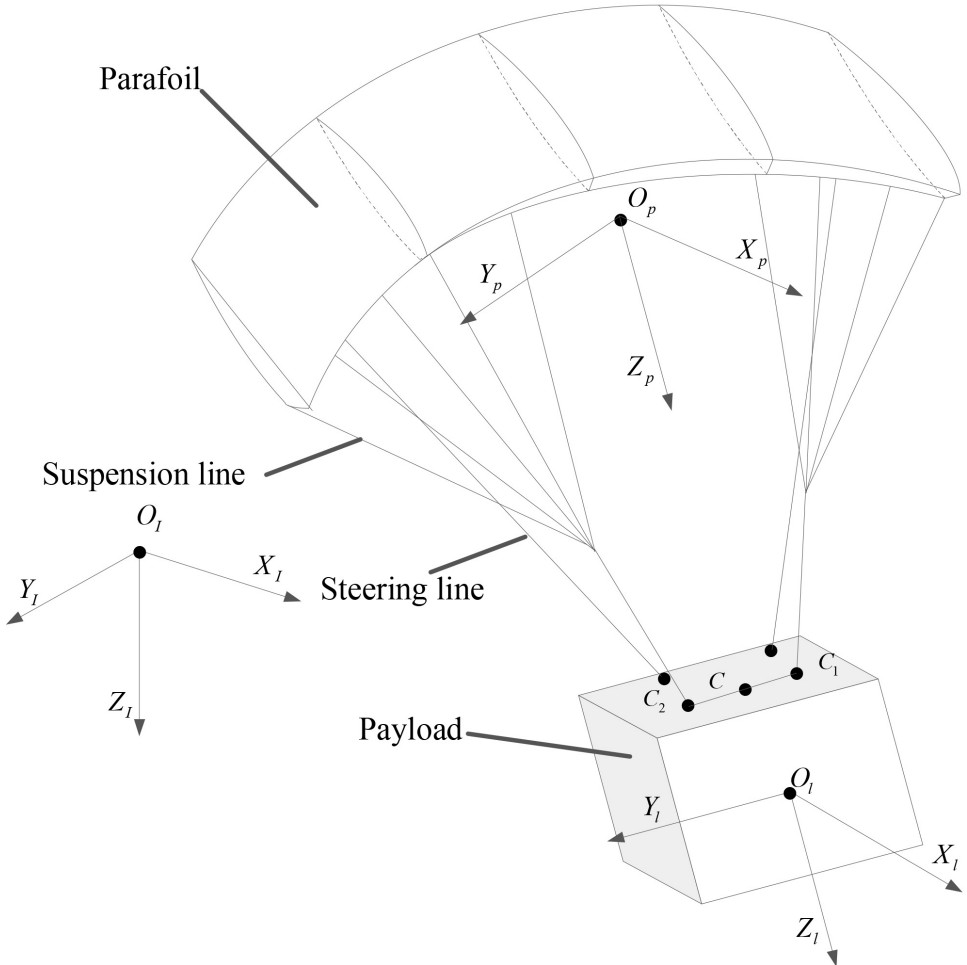

**Figure 1.** The schematic of a parafoil system.

Let $V_p = \begin{bmatrix} v_{x,p} & v_{y,p} & v_{z,p} \end{bmatrix}^T$ and $W_p = \begin{bmatrix} w_{x,p} & w_{y,p} & w_{z,p} \end{bmatrix}^T$ denote the speed and angular velocity of the parafoil, respectively. $V_l = \begin{bmatrix} v_{x,l} & v_{y,l} & v_{z,l} \end{bmatrix}^T$ and $W_l = \begin{bmatrix} w_{x,l} & w_{y,l} & w_{z,l} \end{bmatrix}^T$ denote the speed and angular velocity of the payload, respectively. Using the momentum and momentum moment theorems to analyze the parafoil and the payload, we have

$$\frac{\mathrm{d}T_p}{\mathrm{d}t} + W_p \times T_p = F_p^{aero} + F_p^f + F_p^G + F_p^t \tag{1}$$

$$\begin{aligned} \frac{\mathrm{d}H_p}{\mathrm{d}t} + W_p \times H_p + V_p \times T_p = \\ M_p^{aero} + M_p^f + M_p^G + M_p^t \end{aligned} \tag{2}$$

$$\frac{\mathrm{d}T_l}{\mathrm{d}t} + W_l \times T_l = F_l^{aero} + F_l^t + F_l^G \tag{3}$$

$$\frac{\mathrm{d}H_l}{\mathrm{d}t} + W_l \times H_l = M_l^{aero} + M_l^f + M_l^t \tag{4}$$

where $T$ and $H$ represent the momentum and the momentum moment, respectively. $F$ and $M$ represent force and moment, respectively. Subscripts $p$ and $l$ represent the analysis of the parafoil and the

payload, respectively, and superscripts *aero*, $f$, $G$ and $t$ represent the aerodynamic force, the friction force, the gravity and the tension of suspension lines, respectively.

Taking the apparent mass of the parafoil system into consideration, the momentum $P$ and the momentum moment $H$ of the parafoil and the payload are expressed as

$$\begin{bmatrix} T_p \\ H_p \end{bmatrix} = [A_a + A_r] \begin{bmatrix} V_p \\ W_p \end{bmatrix} \tag{5}$$

$$\begin{bmatrix} T_l \\ H_l \end{bmatrix} = \begin{bmatrix} m_l & 0 \\ 0 & J_l \end{bmatrix} \begin{bmatrix} V_l \\ W_l \end{bmatrix} \tag{6}$$

where $A_a$ and $A_r$ are the real mass and the apparent mass matrix of the parafoil and payload, respectively. $m_l$ and $J_l$ are the mass and moment of inertia of the payload, respectively.

The parafoil and the payload are connected by lines, and the constraint relationships between them are

$$V_l + W_l \times L_{l-c} = V_p + W_p \times L_{p-c} \tag{7}$$

$$W_l = W_p + \tau_s + \kappa_p \tag{8}$$

where $\tau_s$ and $\kappa_p$ denote the relative yaw angle and relative pitch angle of the parafoil and the payload, respectively, which are written in vector forms as $\tau_s = \begin{bmatrix} 0 & 0 & \psi_r \end{bmatrix}^T$ and $\kappa_p = \begin{bmatrix} 0 & \theta_r & 0 \end{bmatrix}^T$. And $L_{l-c}$ and $L_{p-c}$ denote the distance between the center of payload $l$ and the center of parafoil $p$ to the center of the two suspension points $C$, respectively.

Through Equations (1)–(8), the eight-degree-of-freedom dynamic model of the parafoil system can be obtained. For the detailed modeling process, see [2].

## 3. Wind Field Identification

When there is no wind in the environment, the parafoil system is in a gliding flight state under no manipulation or synchronously pulling two steering ropes. When a single steering rope is pulled down at a constant state (also refers to unilateral deflection), the parafoil system is turning to the pulling side. Moreover, in windy conditions, the flight trajectory of the parafoil system drifts with winds, which relates to the speed and direction of the wind. Therefore, it is possible to identify the information of the ambient wind field through the dynamic change of flight trajectories of the parafoil system. In this section, an identification methods of wind fields is investigated [26].

The velocity vector diagram of the parafoil system is shown in Figure 2 [12], which illustrates a vector triangle of wind velocity, constituted by $V_a$, $V_w$ and $V$. $V_a$ denotes the velocity of parafoil systems with respect to the air, named as air velocity. $V_w$ denotes the wind velocity, and $V$ denotes the velocity of parafoil systems with respect to the ground, named as ground velocity. $\beta$ is the sideslip angle, and $\psi$ denotes the yaw angle. $\chi_a$, $\chi_w$ and $\chi$ are angles between $V_a$, $V_w$, $V$ and north direction, respectively.

From Figure 2, the ground velocity $V$ of the parafoil system is equal to the sum vectors of wind velocity $V_w$ and air velocity $V_a$. Assuming that the parafoil system keeps unilateral deflection within a certain time period, and air velocity $V_a$ remain unchanged [1], by the vector triangle, we obtain

$$\begin{cases} \frac{\mathrm{d}x_i}{\mathrm{d}t} = V_{w,x} + V_a \cos(\chi_{a,i}) \\ \frac{\mathrm{d}y_i}{\mathrm{d}t} = V_{w,y} + V_a \sin(\chi_{a,i}) \end{cases} \tag{9}$$

where $i \in \{1, 2, ..., n\}$ denotes for the sampling time sequence. $x$ and $y$ denotes the directions of south-north and east-west, respectively. $\mathrm{d}x_i/\mathrm{d}t$ and $\mathrm{d}y_i/\mathrm{d}t$ denote the ground velocity of the parafoil system in $x$ and $y$ directions at the $i$-th sampling time, respectively. $V_{w,x}$ and $V_{w,y}$ denotes the components of the velocity of horizontal wind (i.e., cross wind) in $x$ and $y$ directions, respectively.

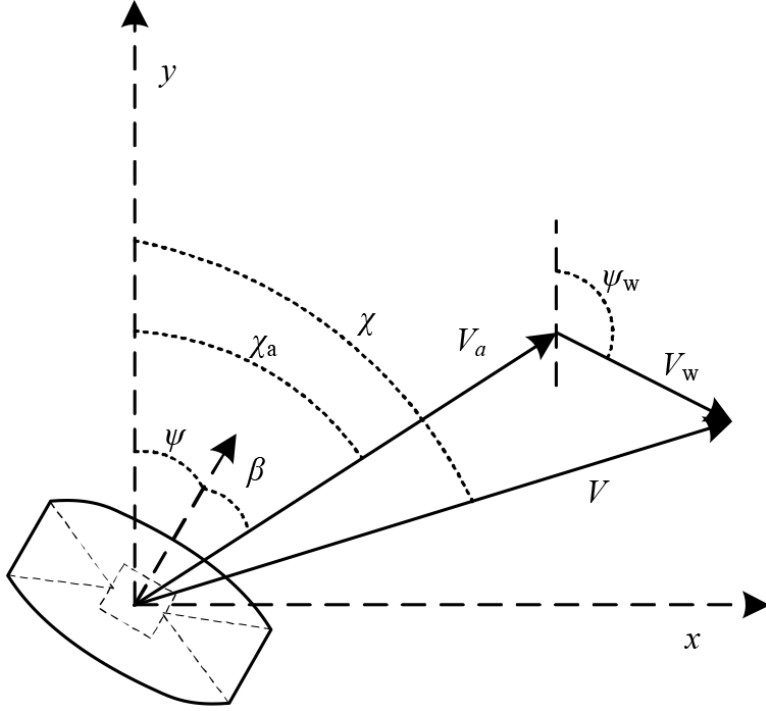

**Figure 2.** Velocity vector diagram of the parafoil system.

Further, Equation (9) is rewritten as

$$
\begin{aligned}
V_a^2 &= (\tfrac{\mathrm{d}x_i}{\mathrm{d}t} - V_{w,x})^2 + (\tfrac{\mathrm{d}y_i}{\mathrm{d}t} - V_{w,y})^2 \\
&= V_i^2 + V_w^2 - 2(\tfrac{\mathrm{d}x_i}{\mathrm{d}t}V_{w,x} + \tfrac{\mathrm{d}y_i}{\mathrm{d}t}V_{w,y})
\end{aligned}
\tag{10}
$$

where $V_i^2 = \left(\tfrac{\mathrm{d}x_i}{\mathrm{d}t}\right)^2 + \left(\tfrac{\mathrm{d}y_i}{\mathrm{d}t}\right)^2$ denotes the ground velocity of the parafoil system at the *i*-th sampling time.

By Equation (10), we get

$$
\begin{aligned}
V_a^2 - E(V_a^2) &= 0 \\
&= V_i^2 - E(V_i^2) - \\
&\quad 2((\tfrac{\mathrm{d}x_i}{\mathrm{d}t} - E(\tfrac{\mathrm{d}x_i}{\mathrm{d}t}))V_{w,x} + (\tfrac{\mathrm{d}y_i}{\mathrm{d}t} - E(\tfrac{\mathrm{d}y_i}{\mathrm{d}t}))V_{w,y})
\end{aligned}
\tag{11}
$$

where $E$ denotes for taking expected value.

For simplicity, let

$$
\begin{aligned}
E(V_i^2) &= \mu_{V2} \\
E(\tfrac{\mathrm{d}x_i}{\mathrm{d}t}) &= \mu_{\dot{x}_i} \\
E(\tfrac{\mathrm{d}y_i}{\mathrm{d}t}) &= \mu_{\dot{y}_i}
\end{aligned}
\tag{12}
$$

Now, the wind field identification of parafoil systems becomes solving linear regression by Equation (13)

$$
\begin{bmatrix}
\tfrac{\mathrm{d}x_1}{\mathrm{d}t} - \mu_{\dot{x}} & \tfrac{\mathrm{d}y_1}{\mathrm{d}t} - \mu_{\dot{y}} \\
\vdots & \vdots \\
\tfrac{\mathrm{d}x_n}{\mathrm{d}t} - \mu_{\dot{x}} & \tfrac{\mathrm{d}y_n}{\mathrm{d}t} - \mu_{\dot{y}}
\end{bmatrix}
\begin{bmatrix}
V_{w,x} \\
V_{w,y}
\end{bmatrix}
= \frac{1}{2}
\begin{bmatrix}
V_1^2 - \mu_{V2} \\
\vdots \\
V_n^2 - \mu_{V2}
\end{bmatrix}
\tag{13}
$$

It is worth mentioning that Equation (13) is not a proper matrix multiplication, and expresses a linear equation set of iterative calculation. It is an equation set of wind field identification when the parafoil system in a turning state. We use the online recursive least square method [27,28] to update

the wind identification results iteratively. Through iterative calculation, $V_{w,x}$ and $V_{w,y}$ are obtained, and consequently, the ambient wind field can be identified.

## 4. Wind Field Prediction

### 4.1. Average Wind Field Prediction Model

Average wind speed over a given altitude band varies with the change of altitudes. The influence of internal viscous friction on the acceleration motion is little in the atmosphere motion process, which can be ignored. The atmospheric dynamics equations [29] in the local vertical, local horizontal coordinates is built as Equations (14)–(17).

The atmosphere motion equation

$$\begin{cases} \frac{du}{dt} = -\frac{1}{\rho}\frac{\partial P}{\partial x} + 2\Omega v \sin\lambda - 2\Omega w \cos\lambda \\ \frac{dv}{dt} = -\frac{1}{\rho}\frac{\partial P}{\partial y} - 2\Omega u \sin\lambda \\ \frac{dw}{dt} = -\frac{1}{\rho}\frac{\partial P}{\partial z} - g + 2\Omega wu \cos\lambda \end{cases} \tag{14}$$

where $u$ and $v$ denote the components of horizontal velocity along $x$ and $y$ axis, respectively. $w$ denotes the vertical velocity along $z$ axis. $\rho$ denotes the density of air. $p$ is the ideal gas pressure. $\Omega$ denotes the angular velocity (with respect to the earth, $\Omega = 15^\circ/s = 7.29 \times 10^{-5}$ rad/s). $\lambda$ denotes the latitude. And $f = 2\Omega \sin\lambda$ denotes Coriolis parameter.

The atmosphere state equation

$$P = \rho RT \tag{15}$$

where $T$ denotes temperature, and $R$ is the gas constant.

The continuous equation

$$\frac{d\rho}{dt} + \rho\nabla\vec{V} = 0 \tag{16}$$

where $\vec{V}$ denotes the three-dimensional wind velocity vector.

The thermodynamic equation

$$\frac{d\varsigma}{dt} = F_\varsigma \tag{17}$$

where $\varsigma$ denotes potential temperature.

For large-scale atmospheric motion systems, the vertical velocity during atmospheric motion is much smaller than the horizontal velocity and the inertial force generated is much smaller than the generated Coriolis force term [30]. The vertical velocity and inertia force can be ignored, and only the horizontal component is taken into account. We obtain

$$\begin{cases} -\frac{1}{\rho}\frac{\partial P}{\partial x} + 2\Omega v \sin\lambda = 0 \\ -\frac{1}{\rho}\frac{\partial P}{\partial y} - 2\Omega u \sin\lambda = 0 \end{cases} \tag{18}$$

Equation (18) shows that the atmosphere reaches to geostrophic equilibrium in large scale motion, which means that Coriolis force and pressure gradient force are at equilibrium state. Plugging in $f = 2\Omega \sin\chi$ into Equation (18) yields to

$$\begin{cases} u_g = -\frac{1}{f\rho}\frac{\partial P}{\partial y} \\ v_g = \frac{1}{f\rho}\frac{\partial P}{\partial x} \end{cases} \tag{19}$$

where $u_g$ and $v_g$ denote two components of geostrophic wind along $x$ and $y$ direction, respectively.

Though the geostrophic wind calculated by Equation (19) is not the actual wind, they are very approximate beyond the area of $\lambda < \pm 15$. When calculating the wind in a free atmosphere, the geostrophic wind can be applied to approximate the actual wind field.

In the friction layer, the pressure gradient force, the turbulent friction, and the Coriolis force are similar [31]. Hence, they cannot be ignored. In the atmospheric boundary layer, the three forces reach to non-geostrophic wind balances

$$
\begin{cases}
-\frac{1}{\rho}\frac{\partial P}{\partial x} + 2\Omega v \sin\lambda + F_x = 0 \\
-\frac{1}{\rho}\frac{\partial P}{\partial y} - 2\Omega u \sin\lambda + F_y = 0
\end{cases}
\tag{20}
$$

The turbulent friction in Equation (20) is calculated by

$$
\begin{cases}
F_x = \frac{\partial}{\partial z}\left(K_m \frac{\partial u}{\partial z}\right) = K_m \frac{\partial^2 u}{\partial z^2} \\
F_y = \frac{\partial}{\partial z}\left(K_m \frac{\partial v}{\partial z}\right) = K_m \frac{\partial^2 v}{\partial z^2}
\end{cases}
\tag{21}
$$

where $K_m$ is the turbulent viscosity coefficient.

Substituting Equation (19) to Equation (20), and assuming that the horizontal pressure does not change with altitude, the horizontal components of average wind is obtained as

$$
\begin{cases}
K_m \frac{\mathrm{d}^2 \bar{u}}{\mathrm{d}z^2} = -f\bar{v} \\
K_m \frac{\mathrm{d}^2 \bar{v}}{\mathrm{d}z^2} = f\bar{u} - fu_g
\end{cases}
\tag{22}
$$

where $\bar{u}$ is the estimation value of $u$, and $\bar{v}$ is the estimation value of $v$.

According to boundary conditions

$$
\begin{aligned}
z \to \infty, \bar{u} = u_g, \bar{v} = 0, \\
z = 0, \bar{u} = 0, \bar{v} = 0.
\end{aligned}
\tag{23}
$$

Through calculating the ordinary differential equation set of Equation (23), we get

$$
\begin{cases}
\bar{u} = u_g[1 - e^{-z/\delta}\cos(z/\delta)] \\
\bar{v} = u_g e^{-z/\delta}\sin(z/\delta)
\end{cases}
\tag{24}
$$

Equation (24) is the predictive model of average wind in the generalized upper friction layer, whose value and angle can be expressed by

$$
\begin{cases}
|\bar{u}| = u_g\sqrt{1 - 2e^{-z/\delta}\cos(z/\delta) + e^{-2z/\delta}} \\
\varphi = \tan^{-1}[e^{-z/\delta}\sin(z/\delta)/(1 - e^{-z/\delta}\cos(z/\delta))
\end{cases}
\tag{25}
$$

where $\varphi$ denotes the angle between the geostrophic wind and average wind. $\delta = \sqrt{2K_m/f} = z_m/\pi$ is Ekman elevation, and $z_m$ is the approximate altitude in boundary layer.

### 4.2. Identification-predictive Method

After the parafoil system being fully deployed in a predetermined airspace, and the GPS module locks the satellite, the wind field identification-prediction procedure starts to identify the wind field at the current altitude and predict the lower altitude. The process is shown in Figure 3. The specific steps are as follows:

1. According to the flight states of parafoil systems, the average wind field $(V_{w,x}, V_{w,y})$ at current altitude is identified by the wind field identification method.
2. Let $(\bar{u}, \bar{v}) = (V_{w,x}, V_{w,y})$, according to Equation (22), $u_g$ can be obtained.
3. Substituting the predicted altitude that to be and $u_g$ into Equation (24), the average wind field $(\bar{u}, \bar{v})$ at the corresponding altitude can be predicted.

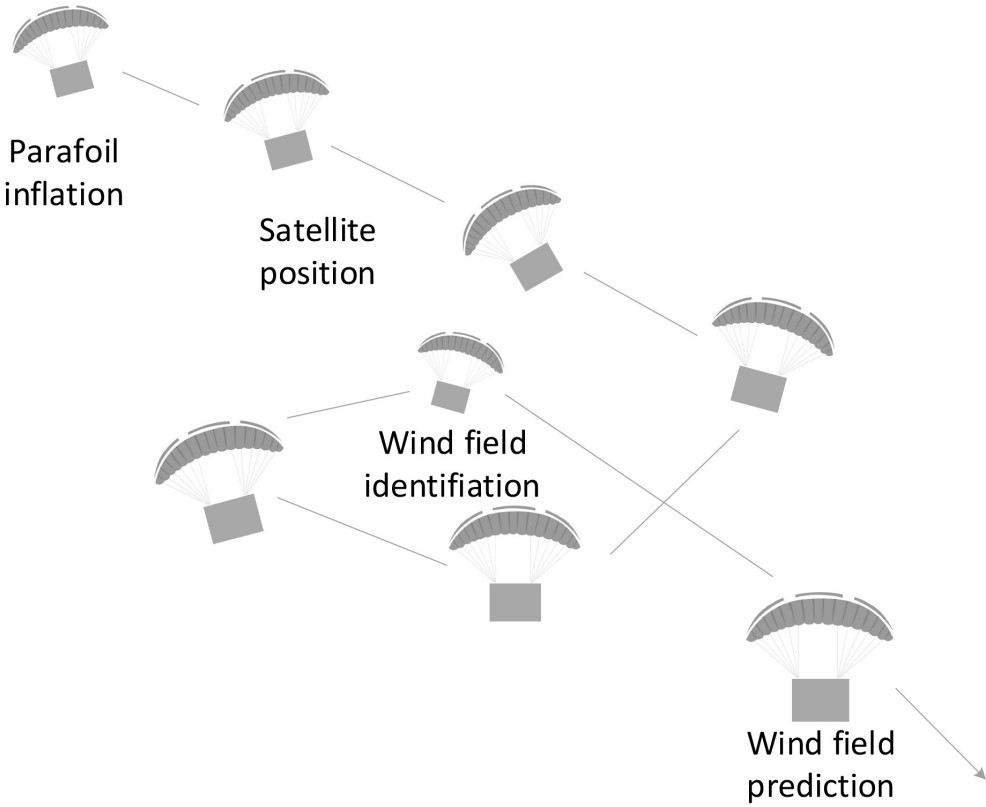

**Figure 3.** The wind field prediction of parafoil systems.

## 5. Simulation and Analysis

### 5.1. Simulation Settings

To verify the proposed methods, simulations of wind field identification and prediction are carried out using PC with Inter(R) Core(TM) i7-6820HQ CPU @2.70GHz and 16.0 GB RAM. The parameters of the parafoil system are listed as Table 1. Assume that the inertial coordinate coincides with the body coordinate at the initial state, and the initial states of the parafoil system are set as follows: velocity $[v_x, v_y, v_z]^T = [16, 0, 2]^T$ (m/s), Eular angles $[\zeta, \theta, \psi]^T = [0, 0, 0]^T$ (rad), and angular velocity $[w_x, w_y, w_z]^T = [0, 0, 0]^T$ (rad/s). The total flight time of the parafoil system is set to 125 s, the crosswind is added at 25 s, the steering rope is pulled down at 37.5 s. The sample interval of parafoil system's position information is set to 1 s, and the position data in 5 s is used to identify the wind field.

**Table 1.** The parameters of the parafoil and the payload.

| Parameter | Value (Unit) |
|---|---|
| Aspect ratio | 1.73 |
| Area of canopy | 22 m$^2$ |
| Length of lines | 3.7 m |
| Rigging angle | 7 ° |
| Length of riser | 0.5 m |
| Mass of payload | 80 kg |
| Characteristic area of drag of payload | 0.5 m$^2$ |

In this paper, the relative error $e_r$ is applied to measure the identification precision, which is defined as

$$e_r = \frac{v_{wset} - v_{wid}}{v_{wset}} \times 100\% \qquad (26)$$

where $v_{wset} = (v_{wset,x}, v_{wset,y})$ is the set wind velocity, and $v_{wid} = (v_{wid,x}, v_{wid,y})$ is the identified wind velocity.

In terms of the special structure of the parafoil canopy, the parafoil system usually flies at a low speed, thus it is vulnerable to wind fields. For simplicity, we only consider the horizontal constant winds in the simulation environment. Different wind fields and deflections are applied, as shown in Table 2.

### 5.2. Simulation of Parafoil System

Figures 4 and 5 show the flight trajectories of the parafoil system without and with winds, respectively (the numbers of trajectories are consistent with the numbers in Table 2). We observe that when the deflections on the parafoil are different, the corresponding trajectories are different. The greater the unilateral deflection is, the smaller the turning radius is and the faster the turning rate is, and vice versa. According to the comparison of trajectories between Figures 4 and 5, the trajectories of the parafoil system drift due to the influence of wind fields. For example, the no.1 trajectory in Figure 4 drifts to the right when adding the lateral wind (2.0 m/s, 4.0 m/s) at 25 s (shown as the no.1 trajectory in Figure 5), and the drifting velocity and direction are related to the wind field.

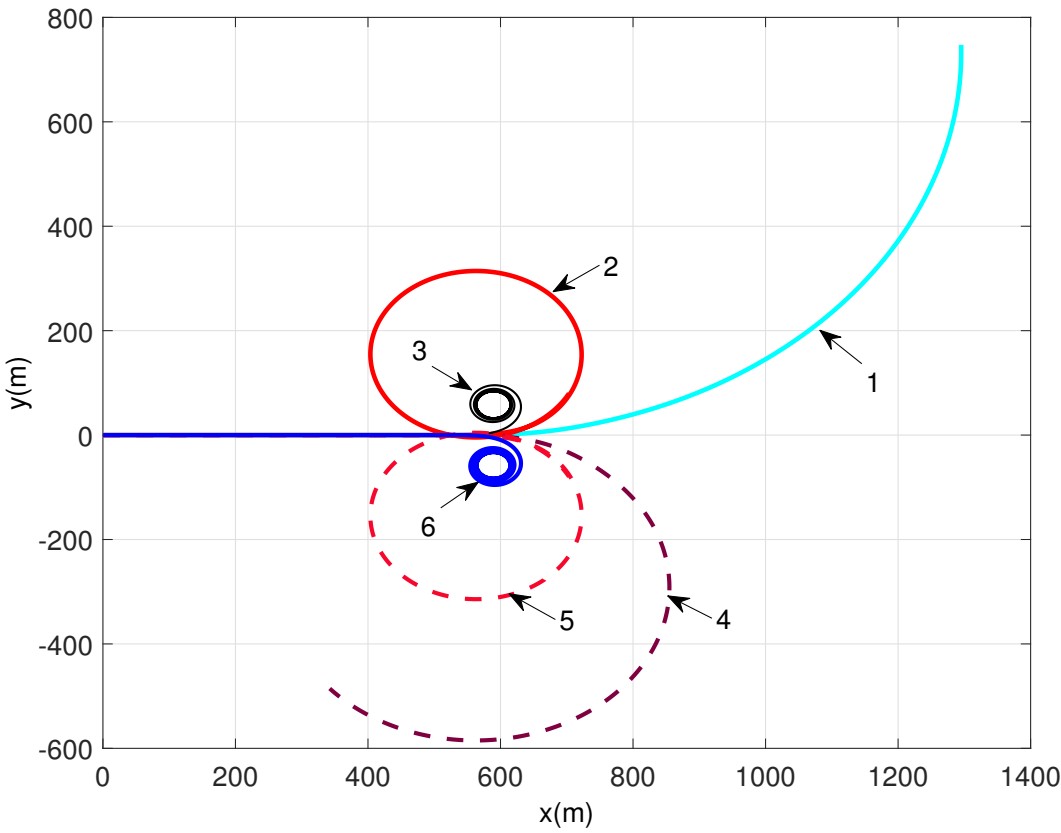

**Figure 4.** Trajectories of the parafoil system without wind.

After adding the wind field $(V_{w,x}, V_{w,y}) = (2.0$ m/s, $4.0$ m/s$)$ and applying 40% unilateral deflection, the simulation results are shown in Figure 6. We observe that the ground speed of the parafoil system changes periodically. When flying against the wind, resistance generates due to the relative motion of the wind, the ground speed of the parafoil system reaches to the minimum. On the other hand, its ground reaches the maximum value in a downwind situation.

As the flight characteristics shown in Figures 4–6, the flight path of the parafoil system contains wind field information. Taken together, these results suggest that it is feasible to identify the wind field by extracting its trajectory changing characteristics.

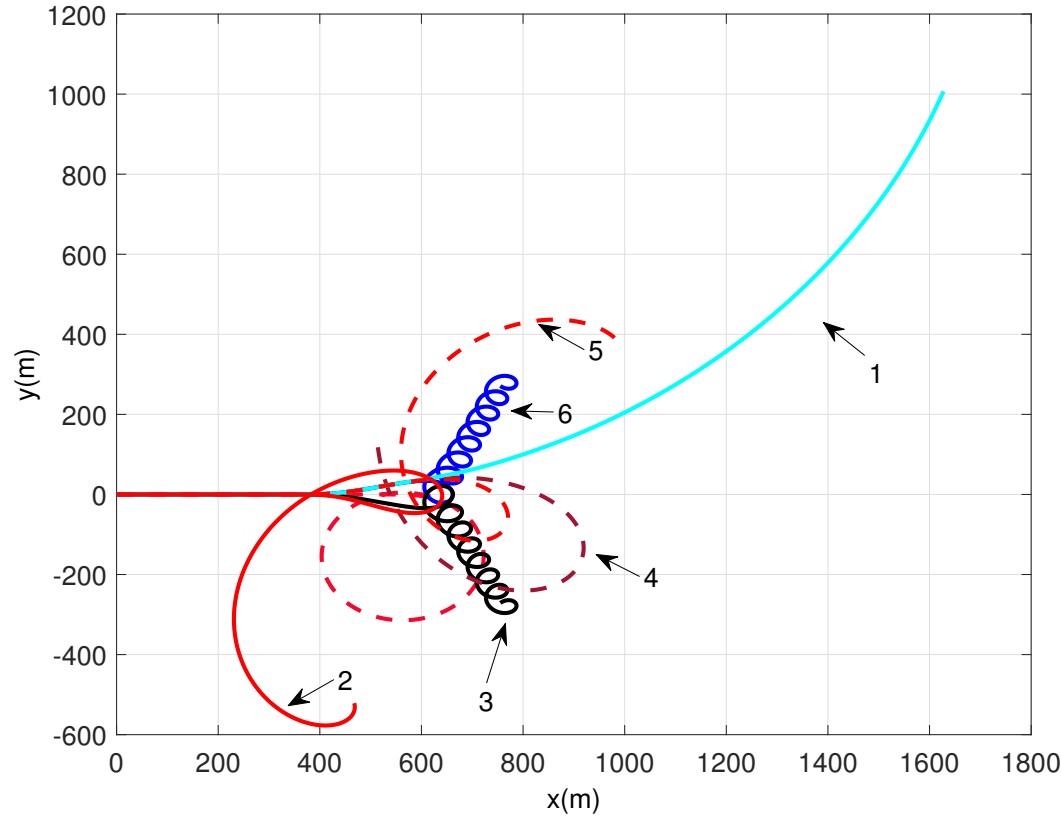

**Figure 5.** Trajectories of the parafoil system with wind fields.

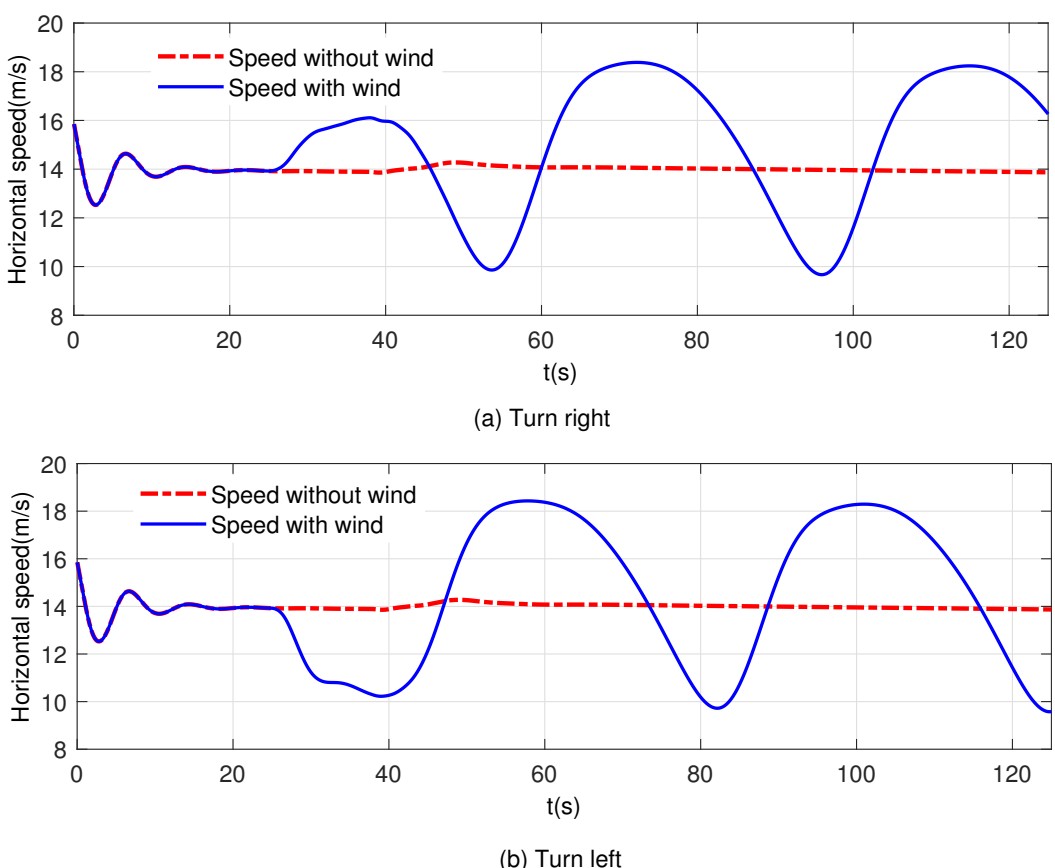

**Figure 6.** Speeds of the parafoil system before and after adding crosswind. (**a**) Turn right. (**b**) Turn left.

### 5.3. Simulation of Wind Field Identification

Under conditions of different deflections and wind fields, wind field identifications are conducted, and the results are shown in Table 2.

**Table 2.** Simulation settings and wind field identification results.

| No. | Deflection (Left, Right) | Wind Velocity Vector $v_{wset}$ (m/s) | Identification Result $v_{wid}$ (m/s) | Relative Error $e_r$ |
|---|---|---|---|---|
| 1 | (10%, 0%) | (2.0, 4.0) | (2.1510, 3.8600) | (7.55%, 3.5%) |
| 2 | (30%, 0%) | (−2.0, −5.0) | (−2.0735, −4.9470) | (3.68%, 1.06%) |
| 3 | (60%, 0%) | (2.0, −4.0) | (1.7385, −4.8084) | (13.07%, 20.21%) |
| 4 | (0%, 20%) | (2.0, 4.0) | (2.0150, 3.9757) | (0.75%, 0.61%) |
| 5 | (0%, 30%) | (2.0, 4.0) | (2.1546, 3.7059) | (7.73%, 7.35%) |
| 6 | (0%, 60%) | (2.0, 4.0) | (1.7385, 4.8084) | (13.08%, 20.21%) |

From Table 2, we observe that the identification results are substantially equal to the applied wind field vectors, which indicates that the proposed method is feasible to identify the ambient wind velocity of parafoil systems. As for practical in-flight identification of the wind field, by calculating the lateral position data collected by GPS, the average wind field information at different altitudes are obtained. According to the relative errors $e_r$, especially from 4 to 6, we can see that with the increase of deflection, the relative errors increase. The best results appear when the deflection is 20%. Thus, to get better wind field identification results, it is best to keep the deflection at 10% to 30%.

Furthermore, we investigate the identification accuracy corresponding to different unilateral deflections. A crosswind with an average speed of 4.472 m/s and a direction of 1.1071 rad is added into the simulation environment, different unilateral deflections of 5%, 10%, 20%, 30%, 40%, 50%, 60%, 70%, 80% are applied to the parafoil system. The results are shown in Figure 7. We see that when the unilateral deflection is too small or large, the wind field identification error is large. As the unilateral deflection is 20%, the speed and direction identification errors are the smallest, i.e., the identification accuracy reaches the highest.

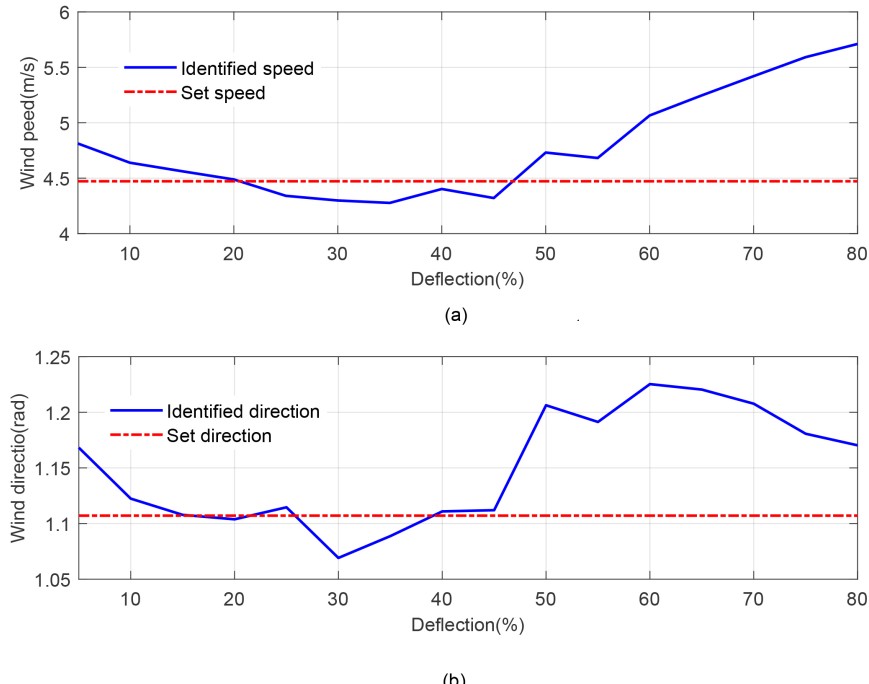

**Figure 7.** Error distribution under unilateral deflection. (**a**) Speed error distribution. (**b**) Direction error distribution.

Furthermore, the same cross wind with the speed of 4.472 m/s and the direction of 1.1071 rad is added in the simulation environment, and 20% unilateral deflection is applied to the parafoil system. The wind field identification results are shown in Figure 8. The results show that after adding of deflection control, the wind field caused significant disturbances to the airspeed of the parafoil system in the early stage when the error of the wind field identification is large. As the flight state of the parafoil system gradually stabilizes, the identification result converges rapidly and enters a stable state after 90 s. The average wind speed identification error is 0.1 m/s, and the wind direction error is close to 0 rad, which indicates high identification accuracy.

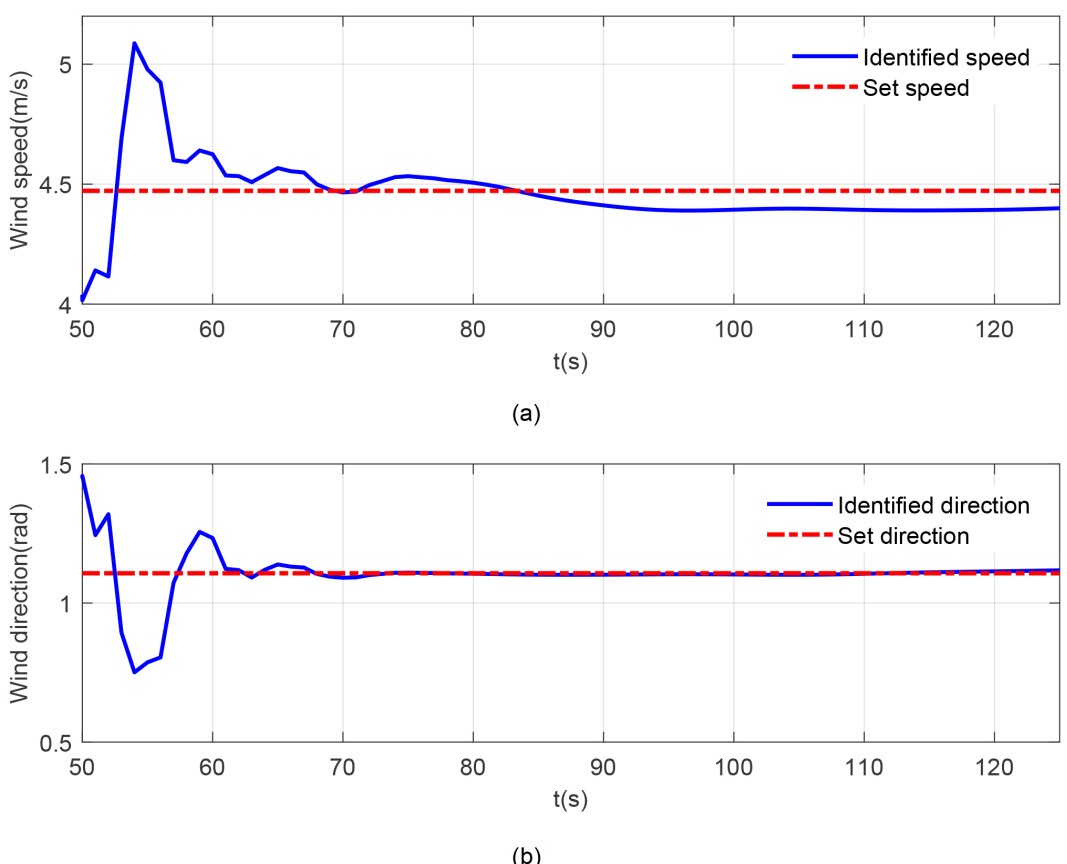

**Figure 8.** Wind field identification results. (**a**) Wind speed. (**b**) Wind direction.

*5.4. Simulation of Wind Field Prediction*

Substituting the identified wind field at a specific attitude into the atmospheric wind field prediction model, the lower average wind field can be predicted to provide a reference for real-time flight path optimization of parafoil systems. Since the meteorological data of the average wind fields at different altitudes are difficult to obtain, in order to facilitate the comparative analysis, this paper selects the measured instantaneous wind velocity of 36.0994°N, 114.9716°E, 12:00, 2 December 2013 (GMT+8) provided by Tianjin Weather Service [32]. The actual measurement wind velocity components are (u = 2.8361 m/s, v = −2.1750 m/s) at the altitude of 1744 m in the 14th layer. We use it instead of the average wind to calculate the geostrophic wind, and get $u_g$ = 2.6581 m/s. Substitute the results into the wind field prediction model, the average wind field at an altitude from 400 m to 1600 m is predicted, the results are as shown in Figure 9.

The comparison results of the predicted wind speed and the actual wind speed are shown in Figure 10. It is observed that the maximum *u*-direction wind speed deviation is 0.7 m/s, while the maximum *v*-direction wind speed deviation is 0.3 m/s. We can conclude that the prediction results can reasonably track the actual wind field despite certain deviations. However, the deviation is not

the exact prediction error, and the actual prediction error is smaller than the deviation. The main reason causing this is that first, the model predicts the average wind field, while the actual wind is the instantaneous wind field data at a specific moment, which is unlikely to be completely consistent. Second, in the atmospheric wind field modelling process, some parameters are idealized. We believe that the prediction results are sufficient to provide adequate wind field information for parafoil systems.

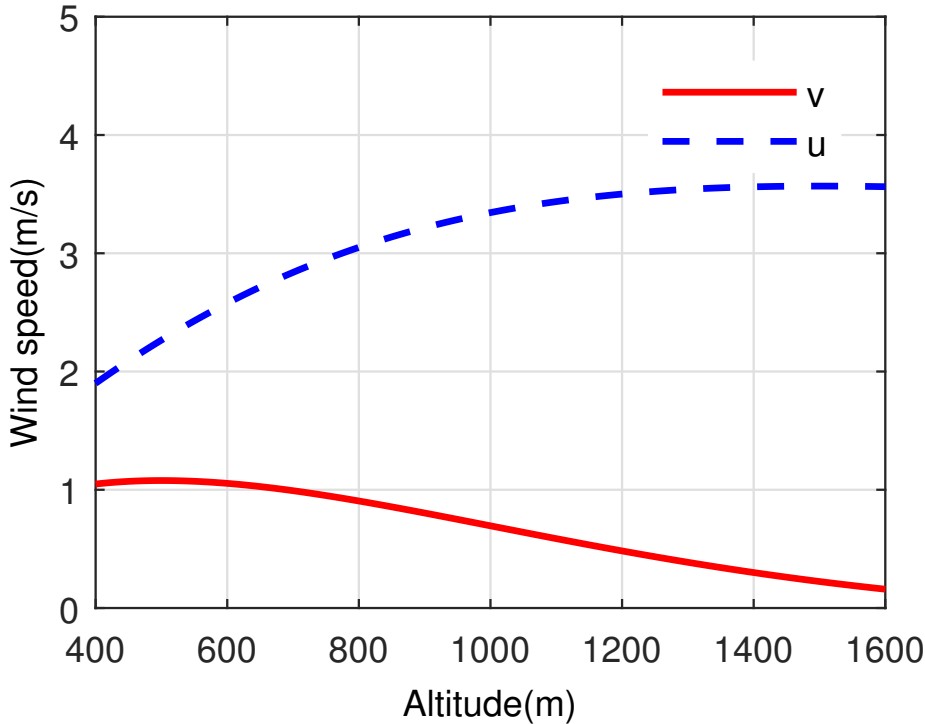

**Figure 9.** Average wind speed prediction results.

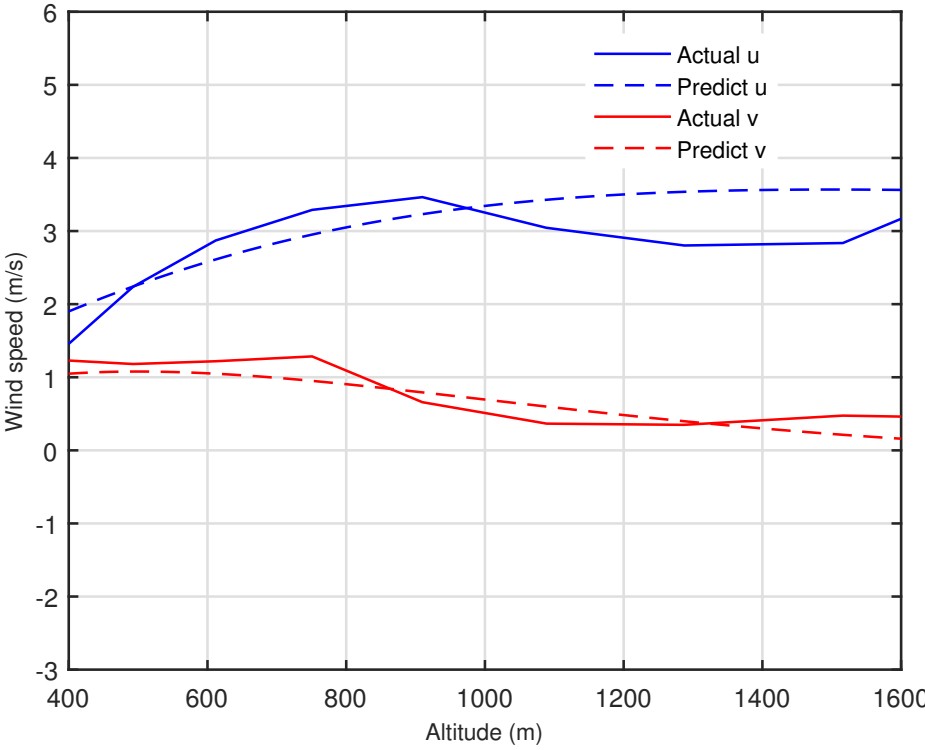

**Figure 10.** Comparison results of actual wind and predicted wind.

## 6. Conclusions

In this paper, we studied a combination method of wind field identification and prediction for autonomous homing of parafoil systems. First, a wind field identification method was studied to identify the ambient wind field according to the flight dynamics of parafoil systems in windy environments. And then, based on atmospheric motion, an average wind field model was proposed to predict the lower-altitude wind field based on the identification results. At last, simulations were conducted to show the effectiveness of the proposed methods. The results show that the average wind field identification method can precisely identify the wind field. Meanwhile, the average wind field prediction model can reasonably predict the average wind field variations at different attitudes. From the practical points of view, the proposed methods provide critical wind information for precise autonomous homing of parafoil systems without velocity measurement sensors that could be difficult to install.

For future work, we will further validate the proposed methods in fight tests of real parafoil systems. Also, we will consider more practical environmental factors, e.g., high-frequency wind fluctuations, and use a general computation intelligence aided design framework for design.

**Author Contributions:** H.G. and J.T. conceived and designed the research; H.G. designed and performed the simulation experiment; M.D. contributed to the theory studies; All authors wrote the manuscript. All authors have read and agreed to the published version of the manuscript.

**Acknowledgments:** This research was funded by the Natural Science Foundation of Anhui Province, China (no.1808085MF183), the University Natural Science Research Key Project Foundation of Anhui Province, China (no. KJ2016A169), the Talents Program Foundation of Anhui Science and Technology University, China (no. ZRC2014444), the National Natural Science Foundation of China (no. 61973172) and the Academy of Finland (no. 315660). The authors would like to express their thanks to Qinsu Han and Panlong Tan for their help with data preparation for this study.

**Conflicts of Interest:** The authors declare no conflict of interest.

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
