# Peer review of "In-flight Wind Field Identification and Prediction of Parafoil Systems"

_applsci, doi:10.3390/app10061958_

Round 1

Author Response

We thank the editors and reviewers for their relevant and useful comments. Your careful reviews and insightful comments have greatly helped us in improving the technical quality and presentation quality of the paper. We have studied these comments carefully and have made the following revisions, which we hope will meet with approval.

In this document, we quote in purple words from the referees’ reports. Our replies follow in black words after “Response”. The modifications in the revised paper are highlighted in red.

Detailed answers to the reviewers’ comments and explanations of the corrections in the revised version of the manuscript are given as follows.

Overall:

- English grammar is rough throughout the paper. Some sentences do not make any sense. Many sentences are missing useful articles. I recommend having the paper edited by a native English speaker.

Response: Thank you for raising this concern. We have this paper edited by a professor that is a native English speaker. We hope the revised version will meet your approval.  

- The paper is not clear on what is novel in this paper

o Wind field identification is not new (and not properly cited!)

ï‚§ Figure 2: Clearly taken from Ref. [10]. Figure uses notation in said reference and not what is used in the paper (V_0 vs. V_a). Equations 9 – 12 are previously included in Ref [10] (partially), the Ph.D dissertation of Ward, and also in Cacan et al. “Autonomous Airdrop Systems Employing Ground Wind Measurements for Improved Landing Accuracy.”

o Wind field prediction at lower altitudes appears to be the main contribution of this paper.

Response: Thank you for your comment. As you suggested, we have added references, such as Cacan et al., and modified the marked position of references. Yes, the main contribution of this paper is the combination method of wind field identification and prediction. We have made revisions to highlight this point.

 - Section 5.4 appears to be the primary validation of the proposed method and is very underwhelming. More data sets need to be shown, both horizontal components (u,v) and vertical components need to be shown and discussed to allow the reader to gauge whether or not this method is valuable. Ideally, flight test data could be included, but I understand that is beyond the scope of this paper.

- Response: Thank you for your comment. As you pointed out, Section 5.4 is the primary validation of the proposed method. We have added comparison results of vertical components v in the revised paper.

Specific (referencing line number):

- 17: “unique construction” -> “slow airspeed in comparison to conventional fixed-wing aircraft”

Response: Thank you for your suggestion. We have revised the revised manuscript.

- 29: Worth citing paper by Cacan et al.

Response: Thank you for your suggestion. We have cited the paper by Cacan as ref[15] in the revised manuscript.

- 38: I do not believe the description of Ref. [21] accurate. This paper proposed an online / in-

flight Monte Carlo approach that conducted path planning that was most robust to a large array of potential wind fields between the currently ID’ed wind and the ground.

Response: Thank you for pointing this out. We carefully read the paper and revised the description in the revised manuscript.

- 55: As noted above, this paragraph needs to clarify the novel contributions of this paper. The ID has been done, whereas the wind prediction appears to be novel.

Response: Thank you for your comments. As you pointed out, the main contribution in this paper is the combination method of online wind identification and prediction. We have made corresponding revisions of this paragraph in the revised manuscript.

- 64: Section ??

Response: Thank you for pointing this out. We have made the revision in the revised manuscript.

- 72: In line vector definitions need a transpose symbol as these are column vectors

Response: Thank you for pointing this out. We have added the transpose symbol in the revised manuscript.

- Equations 1-4: I believe these should be full derivatives, not partial derivatives

Response: Thank you for your comment. As you pointed out, equation 1-4 impartial should be derivative, we have made revisions in the revised manuscript.

- 74: Define superscript G

Response: Thank you for pointing this out. Actually, superscript G denotes gravity, we have explained it in the revised manuscript.

- 82: Are you interested in the centroid of the bodies or the center of mass (payload) and center of pressure (parafoil)?

Response: Thank you for pointing this our. We have revised the expression in the revised manuscript.

- Section 3.1: Needs citations

Response: Thank you for your suggestion. We have used this identification method in our previous research Ref [26], we have cited in the revised manuscript.

- Equation 9: Chi_a -> Chi_a,i

Response: Thank you for pointing this out. We have made revisions in the revised manuscript.

- Section 3.2: This paragraph doesn’t make sense. The accuracy of this method is 100% dependent on accurately knowing the vehicle airspeed V_a. Also, there is no results which show any results of wind field ID when gliding straight. This needs elaboration or must be removed.

Response: Thank you for your concern. We agreed with your comment, and have removed Section 3.2 in the revised paper. 

- 128: Sentence doesn’t make sense…. Average wind speed over a given altitude band?

Response: Thank you for pointing this out. We have revised the sentence as ‘Average wind speed over a given altitude band varies with the change of altitudes’. 

- 131: Justify why the wind field is only estimated down to 400m and not to the ground. Accurate landing of parafoil systems is most dependent on ground wind speeds [Ref. Cacan et al., & Yakimenko et al. “Development andtesting of the miniature aerial delivery system snowflake”]

Response: Thank you for your comments. In this paper, considering that the near-ground wind field, i.e., 0-400m, is greatly affected by the ground environments, we think that the prediction of the average wind field is more suitable for guiding the parafoil system to fly to its landing point in higher airspace. As for near ground precision landing, which has higher requirements for wind field accuracy, Cacan et al [15] provided a better solution. We did not consider this case in this paper.

- 136: I believe lower case p should be upper case P as used in Equation 13

Response: Thank you for your comment. We have changed all small p with capital P, which should be capital P in equation 18-20 (17-19 in old version). In Sec.2, we see the conflict using of capital P, and we changed it to T to donate momentum in the revised paper. 

- 141: “For large scale atmospheric systems… vertical velocity < horizontal velocity” – Perhaps this is a true statement for generic wind field modeling, however, parafoil systems are HEAVILY impacted by local thermals in the wind field which can cause systems to rise in altitude. Better justification of disregarding vertical is needed.

Response: Thank you for your comment. We agree that the parafoil system is heavily impacted by local thermals in the wind field, which may cause rises in its lift force. Here, to simplify equation 14, we disregard vertical velocity based on the justification that horizontal velocity is much larger than vertical velocity.

- Equation 25 / Table 2: Make clear that V_wset = (V_wx, Vwy)_set

Response: Thank you for your comment. We have added explanations after equation 25 in the revised paper.

- 181: Provide further justification or citation on why vehicle is not sensitive to turbulence.

Dryden turbulence models have multiple frequency components which can significantly improve wind field models & impart roll (and turn) the vehicle. Also useful for trying to capture local thermals in the vertical wind field

Response: Thank you for your concern. We are sorry for the inaccurate expression of parafoil system is not sensitive to turbulence. In this manuscript, we only used constant horizontal wind, which is the easiest to apply and justify. As we claimed in conclusion, we will consider a more practical environment, such as wind turbulence, in further work. We have deleted the wrong express in the updated manuscript.

- 194: “Speed” -> ground speed?

Response: Thank you for pointing this out. We have changed ‘speed’ to ‘ground speed’ in the revised manuscript.

- 208-210: This states that greater values of asymmetric deflection does worse, however low asymmetric deflection does not do well. This is later noted in discussion of Fig 8.

Response: Thank you for your concern. We have noticed the inconsistent representation, as you mentioned. We have revised the expression to avoid such conflicts.

- 211: “Decrease of the trajectory data information” doesn’t make sense. The flight time is unchanged and sampling interval is the same, so the data set (amount of information) should be the same… As discussed in Ward ref. [10], part of the reason high deflection does poorly is a result of changing vehicle airspeed.

Response: Thank you for your comment. We have revised thoroughly to avoid such unclear expression in the updated manuscript.

- 224: Flight speed = airspeed?

Response: Thank you for pointing this out. We have made a revision in the updated manuscript.

- 238: Again, unclear why only down to 400m. Ground winds are vital for precision landing.

Response: Thank you for your comments. In this paper, considering that the near-ground wind field, i.e., 0-400m, is greatly affected by the ground environments, we think that the prediction of the average wind field is more suitable for guiding the parafoil system to fly to its landing point in higher airspace. As for near ground precision landing, which has higher requirements for wind field accuracy, Cacan et al [15] provided a better solution. We did not consider this case in this paper.

- 242: Discussion of deviation vs. prediction error isn’t clear.

Response: Thank you for pointing this out. We have given further discussion of the prediction error in the revised manuscript.

- 253: Average wind field ID method is not ‘proposed’ in this work. It is presented based on previous work, assessed, and leveraged to do wind field prediction.

Response: Thank you for your comment. We have revised ‘proposed’ to ‘studied’ in the revised manuscript.

Reviewer 2 Report

Dear Authors,

Thank you for your interesting contribution. I really appreciate the detailed model you created. 

However, I would have several remarks regarding different aspects of your presentation, which you might consider.

Line 16 - the reader may have an impression that references 1-8 cover a wide range of parafoil's different applications, but in fact, more than a half (ref. 5-8) are works by the same set of authors. Please, find references that reflect the text or change the narration. I would suggest amending figure 1 with force and velocity vectors (Pp, Wp, Fp, etc. It will make it easier to follow equations (1 - 6). Line 74 - Could you clarify the meaning of the aerodynamic force and following resistance force. Does this resistance force represent aerodynamic drag? If yes, it is the component of the aerodynamic force. Eq. 8 - How could you add velocities and angles together? Probably, derivatives of angles are missing. Line 97 - Is Va identical with V0 in Figure 2? Eq. 9 - in the second line Vw,x is repeated. Should it be Vw,y? Line 136 - Please be consistent capital P or small p. Line 140 - How the V vector is connected with the velocities in the derivation? Is it a kind of a generic V? Line 152 - Please explain how the horizontal pressure does not change with altitude. The statement is unclear. There is a clear difference between height and altitude. Please be consistent. Conclusions are very limited and do not include many features characteristic to this section. Please, amend it with comparison to the work of other authors. Also, a more detailed description of the future implications might be valuable. Please, perform a proofread of your paper. I spot many typos as well as misuse of several words belonging to specific aerospace terms.

Best regards

Your reviewer.

Author Response

We thank the editors and reviewers for their relevant and useful comments. Your careful reviews and insightful comments have greatly helped us in improving the technical quality and presentation quality of the paper. We have studied these comments carefully and have made the following revisions, which we hope will meet with approval.

In this document, we quote in purple words from the referees’ reports. Our replies follow in black words after “Response”. The modifications in the revised paper are highlighted in red.

Detailed answers to the reviewers’ comments and explanations of the corrections in the revised version of the manuscript are given as follows.

Line 16 - the reader may have an impression that references 1-8 cover a wide range of parafoil's different applications, but in fact, more than a half (ref. 5-8) are works by the same set of authors. Please, find references that reflect the text or change the narration.

Response: Thank you for your comment. We have thoroughly done the literature review and add some recent works. And also, we make revisions in the Introduction to change the references that reflect the narration, we hope the updated version will meet your approval.

I would suggest amending figure 1 with force and velocity vectors (Pp, Wp, Fp, etc. It will make it easier to follow equations (1 - 6).

Response: Thank you for your suggestion. We have amended the figure and added three coordinate systems in the figure, which will make it easier to follow the equations. The original of the coordinate system is the action point of forces.

Line 74 - Could you clarify the meaning of the aerodynamic force and following resistance force. Does this resistance force represent aerodynamic drag? If yes, it is the component of the aerodynamic force.

Response: Thank you for your comment. As a fact, the following resistance force is the friction force, which generates in the connected line with the load when the relative pitching movement between the parafoil and the load occurs. We have made the revision. 

Eq. 8 - How could you add velocities and angles together? Probably, derivatives of angles are missing.

Response: Thank you for your concern. Aa a matter of fact, in equation 8, W denotes the angular velocity, which can be express as (wx wy wz), while τs, and κp denote the yaw angle and relative angle, but written as form of vector, we have revised the expression of τs, and κp in the revised paper.

Line 97 - Is Va identical with V0 in Figure 2?

Response: Thank you for your comment. As you pointed out, it should be Va in Figure 2. We have made the revision in the figure.

Eq. 9 - in the second line Vw,x is repeated. Should it be Vw,y?

Response: Thank you for pointing this out. As you said, it should be Vw,y. We have corrected the typo in equation (9).

Line 136 - Please be consistent capital P or small p.

Response: Thank you for your comment. We have changed all small p with capital P which should be capital P in equation 18-20. In Sec.2, we see the conflict using of capital P, we changed it to T to donate momentum in the revised paper. 

Line 140 - How the V vector is connected with the velocities in the derivation? Is it a kind of a generic V?

Response: Thank you for your comment. As a matter of fact, \vec V is not a generic V, it denotes a three-dimensional wind velocity vector, and the derivation of \vec V is the acceleration of wind.

Line 152 - Please explain how the horizontal pressure does not change with altitude. The statement is unclear.

Response: Thank you for your comment. As you point out, it should be Va in Figure 2. We have made the revision in the figure.

There is a clear difference between height and altitude. Please be consistent.

Response: Thank you for your comment. As you pointed out, the definition of height and altitude is clearly different. We used altitude in the revised paper.

Conclusions are very limited and do not include many features characteristic to this section.

Response: Thank you for your comment. We have rewritten the Conclusion, hope the updated version will meet your approval.

Please, amend it with comparison to the work of other authors.

Response: Thank you for your comment. The main contribution of this paper is the combination method of wind field identification and prediction. We used the identified wind field to predict low altitude wind field. For wind field identification, it is the existing method. For wind field prediction, we have compared the results with the actual wind field, shown as fig. 10. Because the prediction model is built on the specific meteorological data in Tianjin, China, there are no such results in published papers. 

Also, a more detailed description of the future implications might be valuable.

Response: Thank you for your suggestion. We have made revisions in Introduction and Conclusion to highlight future implications. 

Please, perform a proofread of your paper. I spot many typos as well as misuse of several words belonging to specific aerospace terms.

Response: Thank you for pointing this out. We have checked the paper carefully and try out best to correct the typos and misused words. Also, we have this paper edited by a professor that is a native English speaker. We hope the revised version will meet your approval.  

Reviewer 3 Report

In view of this manuscript, I have the following comments:

1.literature review should include more recent works, current literature review in introduction is not supportive.

2. the computation and experiment facility should be given with more details for the numerical simulation and validation.

3. Conclusion and directs for future researches section, has not been properly organised.

4. In the future work, could a general computational intelligence aided design framework be utilised in the smart design process?

5. some typos, such as equation (9)

6. all equations should be numbered, such as E(V2i) = µV2, on page 5.

7. improve quality of figures 3

Author Response

We thank the editors and reviewers for their relevant and useful comments. Your careful reviews and insightful comments have greatly helped us in improving the technical quality and presentation quality of the paper. We have studied these comments carefully and have made the following revisions, which we hope will meet with approval.

In this document, we quote in purple words from the referees’ reports. Our replies follow in black words after “Response”. The modifications in the revised paper are highlighted in red.

Detailed answers to the reviewers’ comments and explanations of the corrections in the revised version of the manuscript are given as follows.

1.literature review should includ6e more recent works, current literature review in introduction is not supportive.

Response: Thank you for your comment. We have thoroughly done the literature review and add some recent works. And also, we make revisions in the Introduction, we hope the updated version will meet your approval.

2. the computation and experiment facility should be given with more details for the numerical simulation and validation.

Response: Thank you for your suggestion. We have added a detailed simulation facility in the revised paper.

3.Conclusion and directs for future researches section, has not been properly organised.

Response: Thank you for your comment. We have reorganized the Conclusion, hope the updated version will meet your approval.

4.In the future work, could a general computational intelligence aided design framework be utilised in the smart design process?

Response: Thank you for your suggestion.  For future work, we will think about using a general computation intelligence aided design framework in design. We have mentioned this in Conclusion in the revised paper.

5.some typos, such as equation (9)

Response: Thank you for pointing this out. We have corrected the typo in equation (9), and we checked the paper carefully and tried out best to correct all typos.

6.all equations should be numbered, such as E(V2i) = µV2, on page 5.

Response: Thank you for your comment. We have numbered the Equation on page 5.

7.improve quality of figures 3

Response: Thank you for pointing this out. We have improved the quality of Figure 3 in the revised version of this paper.

Round 2

Reviewer 1 Report

The paper has been much improved for quality and flow/grammar. The focus of the paper is more clear and the results better show that the method is valuable. 

Small edits:

Line 20: Accurately

Lines 104, 107: ground* velocity (I think it's good to be verbose since you just introduced the different velocities.

Line 161-162: I'm confused between 1) 'sample interval of position information' and 2) 'position data of time span of 5s'. Are you saying that the simulation runs at 1 Hz, but the filter estimates wind speeds at .2 Hz?

Table 2/3: I think both tables should be merged. Would help make results in lines 195-197 more clear.

Line 175: "turning rate is, and* vice versa"

Author Response

We thank the editors and reviewers for their professional work on this paper. We also thank the reviewer for your time and thoughtful comments. We greatly appreciate your help concerning the improvement of this paper.

In this document, we quote in purple words from the referees’ reports. Our replies follow in black words after “Response”. The modifications in the revised paper are highlighted in red.

Detailed answers to the reviewers’ comments and explanations of the corrections in the revised version of the manuscript are given as follows.

The paper has been much improved for quality and flow/grammar. The focus of the paper is more clear and the results better show that the method is valuable.

Small edits:

Line 20: Accurately

Response: Thank you for pointing this out. We have changed ‘Accurate’ to ‘Accurately’ in the revised paper.

Lines 104, 107: ground* velocity (I think it's good to be verbose since you just introduced the different velocities.

Response: Thank you for your suggestion. We have changed ‘velocity‘ to ‘ground velocity’ in the revised paper.

Line 161-162: I'm confused between 1) 'sample interval of position information' and 2) 'position data of time span of 5s'. Are you saying that the simulation runs at 1 Hz, but the filter estimates wind speeds at .2 Hz?

Response: Thank you for raising this concern. In fact,  1) sample interval of position information is the sample time of GPS information of parafoil system in our simulated environment, 2) position data of time span of 5s is used for wind field identification, which means we use 6 position data in 5s to calculated the wind field. The wind estimation also runs at 1Hz, which is the same as the sample rate. We have made further explanations in the revised paper.

Table 2/3: I think both tables should be merged. Would help make results in lines 195-197 more clear.

Response: Thank you for your suggestion.  We have merged Table 2/3 in the revised paper.

Line 175: "turning rate is, and* vice versa" 

Response: Thank you for your comment. We have added ‘and’ in front of ‘vice versa’ in the revised paper.

Reviewer 3 Report

I am happy to accept it.

Author Response

We thank the editors and reviewers for their professional work on this paper. We also thank the reviewer for the acceptance of our paper.